# EEG in Down Syndrome—A Review and Insights into Potential Neural Mechanisms

**DOI:** 10.3390/brainsci14020136

**Published:** 2024-01-27

**Authors:** James Chmiel, Filip Rybakowski, Jerzy Leszek

**Affiliations:** 1Institute of Neurofeedback and tDCS Poland, 70-393 Szczecin, Poland; 2Department and Clinic of Psychiatry, Poznan University of Medical Sciences, 61-701 Poznań, Poland; 3Department and Clinic of Psychiatry, Wrocław Medical University, 54-235 Wrocław, Poland

**Keywords:** EEG, electroencephalography, electroencephalogram, down syndrome, brain oscillations

## Abstract

*Introduction*: Down syndrome (DS) stands out as one of the most prevalent genetic disorders, imposing a significant burden on both society and the healthcare system. Scientists are making efforts to understand the neural mechanisms behind the pathophysiology of this disorder. Among the valuable methods for studying these mechanisms is electroencephalography (EEG), a non-invasive technique that measures the brain’s electrical activity, characterised by its excellent temporal resolution. This review aims to consolidate studies examining EEG usage in individuals with DS. The objective was to identify shared elements of disrupted EEG activity and, crucially, to elucidate the neural mechanisms underpinning these deviations. Searches were conducted on Pubmed/Medline, Research Gate, and Cochrane databases. *Results*: The literature search yielded 17 relevant articles. Despite the significant time span, small sample size, and overall heterogeneity of the included studies, three common features of aberrant EEG activity in people with DS were found. Potential mechanisms for this altered activity were delineated. *Conclusions*: The studies included in this review show altered EEG activity in people with DS compared to the control group. To bolster these current findings, future investigations with larger sample sizes are imperative.

## 1. Introduction

Trisomy 21, the condition characterised by an extra copy of chromosome 21, manifests as the frequent genetic defect known as Down syndrome (DS) [1]. Approximately one in every 700 live births is impacted by it [2], making it one of the most common chromosomal disorders globally [3]. Particular cognitive traits, such as learning challenges, memory problems, and intellectual disabilities, are common in individuals with DS [4]. We are still working to completely understand the complex underlying causes of these cognitive deficits, which involve anomalies in brain connections and neuronal circuits [5].

The imperative to comprehend the neurological basis of cognitive difficulties in individuals with DS underscores the significance of investigating a range of neurophysiological measurements in this population. Understanding the underlying brain processes will be crucial for developing tailored medicines aimed at improving their quality of life. Given its widespread accessibility and non-invasiveness [6], EEG offers a unique opportunity to study the electrical activity and dynamics of the brain in individuals with DS. By monitoring the electrical signals produced by neuronal activity, EEG provides real-time information on neural processing, synchronisation, and connectivity [7]. This information aids researchers in pinpointing critical aspects of brain functioning in individuals with DS.

There are several compelling reasons to conduct EEG research on individuals with DS. Firstly, EEG allows the direct examination of the brain’s electrical activity, facilitating the identification of aberrant neural patterns connected to cognitive impairments. By capturing neural oscillations (resting-state EEG) and event-related potentials (ERPs), EEG can provide light on how the brain functions and reacts to stimuli, which can help explain the cognitive deficits observed in DS patients. Second, its non-invasive and cost-effective nature makes EEG an optimal choice for studying individuals with DS across various age groups, including children and seniors. EEG makes it possible for researchers to look at cognitive functions in individuals with DS throughout their lives, giving important insights into neurodevelopmental trajectories and age-related alterations in brain activity. Comprehending the evolution of cognitive deficits over time and devising tailored therapeutic strategies for particular developmental stages requires an understanding of the long-term viewpoint. Moreover, EEG serves as a valuable technique for examining brain biomarkers associated with cognitive impairments in DS. By examining particular EEG patterns, such as alpha, beta, delta, and theta oscillations, researchers can find potential biomarkers for cognitive deficiencies or risk factors for developing neurodegenerative illnesses such as Alzheimer’s disease, which is more common in individuals with DS [8]. Finding these indicators may aid in early diagnosis and treatment, potentially enhancing outcomes for individuals with DS over the long run.

In this review, we aim to thoroughly assess the existing literature on EEG in people with DS. Given that EEG studies encompass a wide diagnostic spectrum and are categorized into evoked methods, such as ERP for analysing the brain’s response to specific stimuli, and non-evoked methods, such as resting-state EEG for recording the overall electrical activity of the brain, we have specifically chosen one diagnostic method for analysis in this review—resting-state EEG, referred to simply as EEG. This will make the included studies homogeneous and easier to compare. In addition, studies using quantitative QEEG were included in this review. QEEG is an advanced tool that uses EEG technology but adds elements of quantitative analysis, which allows for a more precise diagnosis and planning of individual therapeutic interventions [9]. It is worth adding that EEG power can be measured as absolute power, which refers to the total strength or energy of brain waves within a given frequency range (the results are presented in units of microwatts per square (μV^2^) or another similar power measure), and as relative power, which refers to the proportion of a given frequency range in the total power of brain waves (the results are presented as a percentage of each defined frequency in the overall power of brain waves). The studies included in this review use both methods of measurement. Moreover, EEG recordings should cover moments when the eyes are both open and closed. Proper EEG recordings need to investigate the impact of stimuli on the EEG. A comparison between eyes-open and eyes-closed conditions would be an important evaluative tool. Some rhythms can be obscured by alpha activity and only become evident after the alpha rhythm has been reduced by eye opening [10]. In the included studies, EEG was measured with both eyes open and both eyes closed. Our discussion will encompass various EEG research findings, such as descriptions of brain oscillations and EEG patterns in DS, associations with cognitive function, and the neurodevelopmental alterations noted in a range of age groups. Our objective is to further our understanding of the neurological correlates of cognitive deficits in DS by synthesising this body of research. EEG activity in individuals with DS has been investigated since the 1940s. Gunnarson authored the first book on “imbeciles” in 1945 [11]. Later research on the topic was done, for instance, by Vergani and Aldeghi [12] and by Beley [13]. This earlier research has not been included in the current evaluation because of advancements in the EEG.

## 2. Methods

### 2.1. Data Sources and Search Strategy

For this review, J.C., F.R. and J.L. performed an independent online search using predetermined criteria. The search combined the keywords ‘EEG’ or ‘QEEG’ or ‘electroencephalography’ or ‘electroencephalogram’ with ‘Down syndrome’ or ‘trisomy 21’. Publications within the scope of our search were sourced from the Pubmed/Medline and Research Gate databases, with an access date of October 2023 and publication dates ranging from January 1980 to October 2023.

### 2.2. Study Selection Criteria

Eligibility criteria included clinical studies conducted in English over a specified period that examined EEG in DS. Exclusion criteria were articles not published in English and reviews.

## 3. Results

Figure 1 depicts the screening process as a flow chart. Initially, 5419 records underwent screening, with 5121 being excluded based on the evaluation of their titles and abstracts, primarily due to their topic relevance. Through the search strategies carried out in the database, 254 studies were identified. Of these, 237 studies were excluded on the grounds of their publication type. After a thorough examination of the study abstracts and full texts, 17 articles were considered suitable for inclusion.

### 3.1. Summary and Outcomes of Included Studies

A summary of the studies is provided in Table 1.

#### 3.1.1. Longitudinal Studies

In study [22], 265 individuals with DS, aged between 8 years and 3 months to 55 years and 5 months, participated. Among them, a subset of 28 individuals underwent longitudinal EEG recordings, conducted annually over a span of 8 or 9 years. For one-third of the individuals with DS, a chromosome karyotype was not conducted. Among the remaining two-thirds, the karyotype was typical trisomy, except for four individuals, who exhibited a mosaic type. The initial EEG recording for individuals with DS took place in their twenties for four subjects, in their thirties for 16 subjects, in their forties for seven subjects, and in their fifties for the remaining subjects. During the EEG recordings, the participants closed their eyes. In the occipital area of the brain, the prevalence of the 9 Hz frequency and the range of 10 ± 12 Hz in individuals aged 10 ± 14 years and 15 ± 19 years did not differ significantly from the patterns observed in the healthy control group. In individuals under 9 years old, the predominant component was identified within the 8 and 9 Hz range, with a comparatively lesser presence in the 10 ± 12 Hz range. Between the ages of 10 ± 14 years, the dominance of the theta frequency band diminished, and more than half of the subjects in this age group began to show dominance at 10 ± 12 Hz. However, in contrast to this pattern, the 9 Hz component emerged as the most dominant during the subsequent 20 ± 24 years, and this trend continued until the 35 ± 39-year range. During these time periods, an increasing dominance of the 8 Hz component was also observed, reaching nearly the same prevalence as the 9 Hz component in the 30 ± 34 and 35–39-year ranges. In the subsequent 40 ± 44 years, the prevalence of the 8 Hz component surpassed that of the 9 Hz component, and after the age of 45, it became the most prevalent. For the central area of the brain, the prevalence of the 4 ± 5 Hz range was unusually high in the 20 ± 24-year range, which raised concerns about potential artifacts. Therefore, the analysis focused on higher frequency ranges, excluding 4 ± 5 Hz. The prevalence of 9 Hz was similar to that of 10 ± 12 Hz for individuals aged 10 ± 14 years, and the 9 Hz component became slightly more prevalent in the 15 ± 19-year range. The 9 Hz component reached its peak prevalence in the 25 ± 29-year range but began to decrease thereafter. Conversely, the prevalence of the 8 Hz component exhibited a gradual increase, surpassing other frequencies in prevalence from the age of 30 ± 34 years onward. Similar trends were observed in the frontal EEG, with the prevalence of 9 Hz being notably high in the youngest group but generally consistent with the central EEG in later ages. Among those with DS, the dominant frequency component was 9 Hz or higher for all subjects in their twenties. However, in their thirties, there was an increase in individuals showing frequency components lower than 9 Hz (10 out of 20, 50.0%), and this trend continued in their forties (15 out of 19, 78.9%). Among the subjects in their fifties, three out of four subjects examined (75.0%) displayed this lower frequency pattern. In terms of the age at which the peak frequency in spectra decreased in the follow-up subjects, distinct drops were observed in the thirties for seven individuals and, for two individuals, in the forties. However, a few individuals maintained a dominant frequency of 9 Hz or higher even beyond the age of 45.

#### 3.1.2. Non-Longitudinal Studies

In a research study [14], a cohort of 28 individuals with DS underwent examination. This group comprised of 19 young adults, aged 19 to 37 years, and 9 older adults, aged 42 to 66 years. Additionally, 13 healthy control subjects aged 22 to 38 years were also included in the study. Genetic tests were used to confirm the presence of DS. Four elderly patients had a history of mental disorientation, deterioration, and memory loss and were diagnosed with dementia, but it was not stated which diagnostic tests were used for this purpose. The researchers analysed the EEG readings based on the presence or absence of alpha background activity. A certified electroencephalographer visually interpreted the EEGs. A normal EEG alpha background activity was characterised by symmetrical and regular rhythms (8.5 to 13 Hz) at the back of the brain, particularly noticeable when participants closed their eyes and diminishing upon eye opening. Abnormal background activity was defined as either the absence of an alpha background or an irregular, low-amplitude, mixed-frequency (6 to 11 Hz) background. Among the 19 young adult DS patients, thirteen (68.4%) exhibited normal alpha background activity, while six (31.6%) showed abnormal activity. In the group of nine older DS patients, five (55.6%) had a normal alpha background activity, and four (44.4%) had abnormal activity. Some of the older DS patients with abnormal EEG readings displayed certain neurological patterns, such as frontal slowing or sharp transients in specific brain regions. Among the older DS patients who showed a loss of alpha background in their EEGs, all exhibited signs of dementia, with two confirmed cases of Alzheimer’s disease upon postmortem examination. The study also explored the cognitive aspects of these participants. In the young adult DS patients, there were no significant differences in neuropsychological findings between those with normal EEG backgrounds and those with slower EEG backgrounds. However, among the older DS patients, those with abnormal EEG backgrounds exhibited notable impairments in visuospatial skills and attention compared to those with a normal alpha background. Additionally, these patients exhibited a non-significant decline of at least 50% in language function, one attention test, and the test of visual recognition memory when compared to older patients with normal alpha background activity.

In the research [15], a group of 32 individuals diagnosed with DS, ranging in age from 21 to 60, were the subjects of study. There was no information available regarding whether DS was confirmed by genetic tests. There was a wide range of variability in mental deficiency, with standard deviation values being significantly larger than the means. It was not reported how many patients had significant cognitive decline or confirmed dementia. Cognitive functions were measured via a battery of neuropsychological tests and the mini-mental state (MME) examination. The researchers recorded EEG and QEEG measurements on two occasions. Initially, measurements were taken while the participants were in a relaxed, awake state, first with their eyes closed and then again with their eyes open. They focused on analysing the absolute amplitude and power of different frequency bands: delta (1.46–3.91 Hz), theta (4.15–7.32 Hz), alpha (7.57–13.92 Hz), and beta (14.16–20.02 Hz)—as well as the peak frequency (Fp) and mean frequency (Fm, ranging from 1.46 to 20.02 Hz). Their analysis of the conventional EEG readings showed that 16 patients had a normal background rhythm in the occipital region, while 12 showed mild slowing, 3 exhibited moderate slowing, and only 1 patient displayed marked slowing. Comparing the DS patients to controls, they found that the levels of delta, theta, alpha, and beta activity were higher in the DS group. In terms of the alpha band specifically, the researchers discovered that the amplitude ratio between EC and EO was significantly reduced in the DS patients when compared to the control group. Furthermore, the study showed significant connections between various neuropsychological measures, the mini-mental state examination (MMS), the Blessed scale, and the pooled alpha EC/EO ratio. A positive correlation was observed, indicating that a higher alpha EC/EO ratio was associated with better cognitive performance. Additionally, the alpha EC/EO ratio showed an inverse correlation with the age of the patients, suggesting that older patients tended to have a lower ratio.

In this study [16], the EEG data from a pilot investigation were focused on 38 adolescents with DS, including 20 males, with an average age of 18.7 years. DS was confirmed by genetic tests. They were compared to a group of 17 young individuals without DS who were of similar ages. The research setup involved placing eight scalp electrodes at specific positions: Fp1, Fp2, C3, C4, T3, T4, O1, and O2. During the recordings, the participants were instructed to keep their eyes closed. The study concentrated on analysing specific EEG rhythms, namely delta (2–4 Hz), theta (4–8 Hz), alpha 1 (8–10.5 Hz), alpha 2 (10.5–13 Hz), beta 1 (13–20 Hz), and beta 2 (20–30 Hz). The results highlighted distinct patterns within the DS group. Notably, there was a noticeable increase in power density values in the delta frequency band across the frontal areas of the scalp. Additionally, the DS group exhibited reduced power density values in the alpha, beta, and gamma frequency bands.

The research outlined in the study [17] involved enlisting 45 individuals with DS. EEG data were collected from these individuals while they were in a resting state upon waking, with their eyes closed. DS was confirmed by genetic tests. All participants were categorised according to the diagnostic criteria suggested by the AAMR-IASSID working group for diagnosing dementia in individuals with developmental disability. All individuals recruited for this study were confirmed to be free from dementia. The EEG readings were taken from 19 specific electrode positions on the scalp, including areas such as Fp1, Fp2, F7, F3, Fz, F4, F8, T3, C3, Cz, C4, T4, T5, P3, Pz, P4, T6, O1, and O2. The focus of the analysis was on different frequency bands, including delta (2–4 Hz), theta (4–8 Hz), alpha 1 (8–10.5 Hz), alpha 2 (10.5–13 Hz), beta 1 (13–20 Hz), and beta 2 (20–30 Hz). When compared to a control group of individuals without DS, the DS group exhibited certain distinct patterns. Specifically, the amplitude of sources within the alpha 1 and alpha 2 frequency bands was reduced in the DS group, whereas there was an increase in the delta band activity. Further analysis, carried out after the initial findings, revealed that the pattern of sources in the control group was higher compared to the DS group. This difference was evident in the LORETA solutions, which highlighted specific brain regions. For instance, the alpha 1 and alpha 2 sources in the central, parietal, occipital, temporal, and limbic regions were more pronounced in the control group. Similarly, in beta 1 sources, the control group displayed heightened activity in parietal, occipital, temporal, and limbic areas. On the other hand, an opposing trend was observed in occipital delta sources, where the DS group exhibited different patterns compared to the control group.

In the study [18], a cohort of 88 children with DS, aged between six months and five years old, were compared to a larger group of 277 typically developing children. There was no information available regarding whether DS was confirmed by genetic tests. EEG recordings were taken from specific electrode positions on the scalp, including F4-C4, F3-C3, T4-T6, T3-T5, P4-O2, and P3-O1, while the children were awake, with their eyes open. The data analysis involved segmenting the spectra from the fronto-central and parieto-occipital leads into six distinct frequency bands: K1 for subdelta (0.4–1.5 Hz), K2 for delta (1.5–3.5 Hz), K3 for theta (3.5–7.5 Hz), K4 for alpha (7.5–12.5 Hz), K5 for beta 1 (12.5–19.5 Hz), and K6 for beta 2 (19.5–25 Hz). The study measured absolute and relative power. When observing EEG development across the fronto-central region, younger children with DS showed relatively minor differences in subdelta, delta, and theta bands compared to their typically developing peers. However, with increasing age, these children displayed an increasing amount of theta activity, reaching its peak at around two years of age. The most noteworthy deviations from normal EEG development were observed in the alpha and beta 1 bands. Up to their first year of life, the average alpha band values in DS EEGs were within one standard deviation of the control group. Between the ages of two to five years, a distinct reduction in alpha activity was evident in children with DS, particularly notable at ages two and three. In the parieto-occipital area, there was a gradual increase in relative subdelta, delta, and theta activity with age, marked by a rise in theta activity at two years of age and increased subdelta and delta activity at five years. Similar to the frontal region, deviations in the alpha and beta 1 bands were observed, becoming more pronounced as the children with DS grew older. When comparing the frontal and posterior regions, a similar pattern emerged, where the reduction in relative alpha and beta power became more prominent as age increased. While the development of low frequencies was clear, the older children with DS showed a relative increase in these low frequencies. Regarding the absolute power of different frequency bands, there were several findings. Over the fronto-central area, subdelta, delta, theta, and beta 2 power were higher in DS children, while the absolute alpha power showed a deficit. Similarly, in the parieto-occipital area, the absolute power of different frequency ranges was higher in DS children, particularly due to increased subdelta, delta, and theta power. It’s also noteworthy that, in some instances, the absolute alpha power in DS children was higher up to the age of three years but reduced in four- and five-year-olds. Higher frequency bands such as beta 1 and beta 2 also showed a significant increase in absolute power.

In study [19], a cohort of 40 individuals aged 15 to 54 with DS underwent examination. There was no information available regarding whether DS was confirmed by genetic tests. The study focused on analysing EEGs from specific locations on the left hemisphere (F3-A1, C3-A1, P3-A1, O1-A1). Two primary aspects were investigated: (1) determining the highest frequency point of occipital alpha rhythms and (2) assessing relative power, which involved calculating relative power values within the frequency range of 4–30 Hz. EEGs from all four locations were digitally recorded, and relative power was calculated across six distinct frequency bands: theta1 (4 Hz to <6 Hz), theta2 (6 Hz to <8 Hz), alpha1 (8 Hz to <10.5 Hz), alpha2 (10.5 Hz to <13 Hz), beta1 (13 Hz to <20 Hz), and beta2 (20 Hz to <30 Hz). The study found a significant inverse relationship between the peak frequencies of alpha rhythms in the left occipital location (O1-A1) and the chronological age of individuals in the DS group. The comparison with a control group revealed several age-related differences in relative power. In the youngest age group of DS individuals, there were significant increases in the relative power of the theta1 and beta2 bands across all locations, as well as an increase in the beta1 band in the F3 and C3 locations. The 25–34 age group also displayed increased relative power in the beta1 and beta2 bands. Among the DS 35–44 age group, notable increases in the relative power of the theta2 and alpha1 bands were observed. Individuals aged 45–54 showed an increased relative power in the theta2 and alpha1 bands, particularly in the occipital EEG. Across all age groups of DS, the relative power in the alpha2 band was consistently lower compared to the control group, especially in the posterior leads.

In the research [20], 31 individuals with DS participated. There was no information available regarding whether DS was confirmed by genetic tests. The individuals with DS exhibited cognitive impairment, as assessed through MMS scores, but there was no information available to suggest that they had been diagnosed with dementia. The study included EEG recordings taken from specific electrode positions, namely, T6-O2, T5-O1, C4-P4, C3-P3, F4-C4, and F3-C3. Data taken from T6-O2 was the primary focus. During the recordings, the participants kept their eyes closed. Various parameters were quantitatively analysed, and the absolute amplitude and power of specific frequency bands—delta (1.46–3.91 Hz), theta (4.15–7.32 Hz), alpha (7.57–13.92 Hz), and beta (14.16–20.02 Hz)—were calculated. Additionally, peak frequency and mean frequency values within the range of 1.46–20.02 Hz and for the combined alpha and theta range of 4.15–13.92 Hz were determined. The findings highlighted significantly higher levels of slow wave activity, particularly in the theta and delta bands among the DS patients. Delta amplitude and peak frequency were specifically chosen among various measures to gauge EEG slowing and its correlation with cognitive performance in the participants. To better understand the impact of age on these EEG patterns, the DS patients were divided into two groups: those under 40 years old (17 individuals) and those aged 40 and older (10 individuals). The older patients exhibited a lower peak frequency (8.3 ± 0.8 Hz vs. 9.0 ± 0.7 Hz) and mean frequency (8.6 ± 0.5 vs. 9.0 ± 0.4 Hz) within the 4.15–13.92 Hz range compared to the younger group. Age consistently correlated with frequency parameters, across the entire DS group and specifically in the DS patients aged 40 and above. Moreover, within the DS group, peak frequency demonstrated a correlation with cognitive scores (measured by the MMS score) and functions related to vision, motor skills (praxis), speech, and list learning. Additionally, significant correlations were observed between delta amplitude and praxis functions, as well as list learning in DS patients.

In the research [21], 32 adults with DS, (15 in their 20s, 9 in their 30s, and 8 in their 40s) underwent EEG recordings. Ninety percent of patients had genetically confirmed trisomy 21, and 10% of patients had the mosaic type. It was not reported whether any of the patients had been diagnosed with dementia, but the test results indicated cognitive decline. The recordings were conducted using a referential montage with 16 scalp electrodes placed at specific locations (Fp1, Fp2, F3, F4, F7, F8, Fz, C3, C4, P3, P4, Pz, T5, T6, O1, and O2). The study aimed to assess changes in the dominant basic brain activity by calculating the mean frequency of the left occipital region (O1) based on data points within the theta and alpha frequency bands (4 to 13 Hz). Comparisons between the three age groups of DS individuals revealed increased theta 1 activity at O1 and increased theta 3 activity at seven sites, predominantly in the frontal and parietal regions among individuals in their 40s compared to those in their 20s. Additionally, alpha 4 and beta 1 activities were generally reduced across 11 and 8 sites, respectively. Upon closer examination, comparing the 30s group to the 20s group showed a higher occurrence of delta activity at one site (Fp2), and a decrease in alpha 3 activity at one site (F3), while alpha 4 activity was diminished at six sites, predominantly in frontal and parietal regions. When comparing the 40s and 30s groups, few sporadic differences were observed in certain sites and frequency bands. Specifically, theta 1 activity was more frequent at one site (O1), theta 3 was more frequent at two sites (F4, P4), and alpha 4 activity was less frequent at one site (P4). The study also revealed a decreasing trend in mean frequency with advancing age within the DS group, with values of 9.37 Hz for the 20s group, 9.17 Hz for the 30s group, and 8.76 Hz for the 40s group. A notable difference was found between the 20s and 40s age groups.

In the study [23] involving individuals with DS, the participants closed their eyes during the EEG recordings. All participants included in the study had their trisomy 21 genetically confirmed. Cognitive decline was assessed using the Cambridge examination of mental disorders of older people with Down syndrome and others with intellectual disabilities (CAMDEX-DS). To be eligible for the study, participants were required to show no signs of decline in accordance with this questionnaire. The researchers collected measurements of absolute and relative power in specific frequency bands (delta 0.5–4 Hz, theta 4–8 Hz, alpha 8–13 Hz, beta 13–30 Hz) for different brain regions (frontal and occipital). They also calculated features related to the highest point of the alpha wave, which is defined as the frequency with the peak strength within the 8–13 Hz range. When analysing the EEG data from the occipital region, they found notable differences between those with DS and typical development (TD) controls. The individuals with DS exhibited significantly higher absolute and relative power in the delta band, along with relatively higher power in the theta band. On the other hand, they showed significantly lower absolute and relative power in both the alpha and beta bands. Additionally, the alpha peak amplitude was notably lower in the DS group. The largest differences were seen in relative alpha power, accounting for 56.5% of the variation between the two groups. Similar findings were observed in the frontal region, where all group differences (including both absolute and relative values) were statistically significant, except for the absolute alpha and beta power and alpha peak frequency. Overall, in the frontal region, individuals with DS had notably higher absolute and relative power values in the delta and theta bands. Conversely, relative power values in the alpha and beta bands, along with both absolute and relative alpha peak amplitudes, were considerably lower. The most significant differences were noted in the absolute alpha peak amplitude and relative alpha power, with these variables accounting for 28.8% and 20.9% of the group variation, respectively.

In study [24], all participants had the classical phenotype of DS. All included patients were without dementia, based on meeting several requirements, including participation in social activities, the absence of clinical evidence of cognitive decline, absence of spatial disorientation, and others. The researchers conducted their analysis using the O2-A2 channel while the participants had their eyes closed. They categorised EEG frequencies into different bands: delta (0.4–3.9 Hz), theta (4.0–7.9 Hz), alpha (8.0–12.9 Hz), beta (13.0–17.9 Hz), and gamma (18–50 Hz). The results indicated that individuals with DS exhibited higher absolute power in the delta, theta, and beta bands, as well as in the total power of the resting EEG. When comparing the absolute and relative powers of each specific frequency range between DS subjects and control subjects, those with DS showed significantly higher values in the 4.5 Hz and 8.8 Hz frequency ranges. In relation to cognitive abnormalities, there were significant negative correlations between resting power in the 4.5 Hz and 8.8 Hz frequency ranges and cognitive test performance. Moreover, there was a strong positive correlation between the MMSE and PAT scores.

The research [25] examined the EEGs of six individuals with DS, aged between 24 and 78 years old and with IQs ranging from 6 and 24. There was no information available regarding whether DS had been confirmed through genetic tests. EEG data from 19 scalp electrodes were collected using referential and bipolar montages over a 6 s period. Various aspects of the EEG, including univariate, bivariate, and multivariate measures, were measured and compared to normative values adjusted for age. The participants kept their eyes closed during the study. The findings revealed an increase in both absolute and relative power in the delta and theta frequency bands, accompanied by a simultaneous decrease in alpha and beta activity.

The research [26] enrolled 36 individuals with DS, and their EEG data, specifically recorded from the left occipital lead (O1-A1), were examined. DS was confirmed by genetic tests. The study involved calculating peak frequency and relative powers in standard EEG bands, with alpha further divided into 8–10.5 and 10.5–13 Hz ranges. Eight additional participants with DS were excluded due to an inability to detect the alpha rhythm. To explore age-related changes, the subjects were categorised into four age groups: (1) 15–24, (2) 25–34, (3) 35–44, and (4) 45–54 years old. The participants closed their eyes during the EEG recording. The DS participants were compared to two control groups: 47 healthy volunteers (control) and 42 individuals with intellectual disabilities without DS (MR). The results revealed a significant negative correlation between the alpha peak frequency and age, following a linear pattern—a relationship not observed in either control group. Notable differences in the alpha frequency compared to controls were evident across all age groups, including the 15–24 age range. In comparison to healthy controls, individuals with DS exhibited no differences in beta and a significant decrease in relative power in alpha2 for ages 25–54. Older adults (35–54 years) also demonstrated a significant increase in theta2 and alpha1. Increases in theta1 were observed in the 15–24 and 35–44 age groups. When compared to MR controls, similar differences in alpha1 and alpha2 were noted in older adults.

The research [27] involved participants with DS, including 33 young adults (20–35 years) and 12 older adults (36–55 years), as well as healthy controls in the corresponding age groups. There was no information available regarding whether DS was confirmed by genetic tests. The participants kept their eyes closed during the QEEG. The main goal was to investigate the relative and absolute power of delta, theta, alpha (8–11.9 Hz), and beta bands, along with the topographical distribution of each. Additionally, participants underwent various neuropsychological tests: abstract reasoning (Raven’s colour matrices; scores also transformed to give an IQ score), language comprehension (Tolken test), and language production (verbal fluency). Dementia was assessed using the Shapiro criteria; some patients had dementia. The results revealed abnormal EEGs in 73% of young adults with DS and 92% of older adults with DS. Both young and older adults with DS exhibited increased power in delta (both relative and absolute, predominantly in the centro-anterior regions), theta (both absolute and relative, mainly in the centro-posterior regions), and beta (both absolute and relative, mostly in the parieto-temporal regions), along with a decrease in alpha. The decrease in alpha was statistically significant only in the posterior regions among older adults. When comparing older adults with DS to young adults with DS, only the older adults showed a significant increase in theta power. Overall, participants with DS had a significantly slower alpha peak rhythm compared to controls (9.2 ± 1 Hz vs. 9.7 ± 0.3 Hz), with no correlation between the alpha peak frequency and age in either group. In the participants with DS, the alpha peak frequency positively correlated with scores on neuropsychological tests such as Raven’s colour matrices, Rivermead behavioural memory test, and Tolken test. The severity of cognitive impairment was associated with a higher prevalence of abnormalities, including decreased alpha power, increased theta in the posterior regions, and decreased delta across the scalp. This pattern was more pronounced in participants with dementia.

The study [28] involved 45 participants with DS. There is no information available regarding whether DS was confirmed by genetic tests. Participants had their eyes closed during the recordings. The participants were divided into two age groups: 33 individuals aged 20 to 35 and 12 individuals aged 36 to 56. All participants underwent QEEG, and the relative and absolute power of delta, theta, alpha (8–11.9 Hz), and beta were analysed. Neuropsychological tests were also conducted, assessing abstract reasoning (RCM), language comprehension (TT), production (VF), attention (CT), and memory (RBMT). RCM scores were transformed to represent IQ. Dementia was diagnosed through clinical assessment using the Shapiro criteria and cognitive test outcomes. The results indicated that 73% of young adults and 92% of older adults had “abnormal” EEG readings, with a significant increase in delta (75%), theta (37%), and beta (37%) power in the younger adults’ group, along with a 58% decrease in alpha power. In the older adults’ group, there was a significant increase in delta (54%), theta (91%), and beta (36%) power, with a 45% decrease in alpha power. Topographically, there was a significant increase in delta power in the centro-anterior and parieto-temporal regions, and a significant decrease in alpha power over the posterior regions in the older adults’ group. Additionally, there was a significant increase in absolute and relative theta power over the centro-posterior regions in the older adults’ group. Cognitive impairment was associated with a decrease in alpha power, an increase in theta power (particularly in the posterior regions), and an increase in delta power across the scalp. This pattern was more pronounced in participants with DS and dementia. It is worth noting that individuals with DS without EEG abnormalities performed significantly better on most measures compared to those with abnormalities.

The study [29] involved 25 adult participants with DS. There is no information available regarding whether DS was confirmed through genetic tests. Participants had their eyes closed during the recordings. Participants performed a series of neuropsychological tests: MMSE, Wechsler adult intelligence scale, attentive matrices test, digit span test, Rivermead behavioural memory, short story recall, Raven’s coloured progressive matrices, Tolken test, semantic verbal fluency, and geometric shape copy. The EEG data were compared with that of control subjects who were matched in terms of age and gender. This comparison involved looking at absolute and relative power in specific frequency bands, including delta (1–3 Hz), theta (4–7 Hz), alpha1 (8–9 Hz), alpha2 (10–12 Hz), beta1 (13–18 Hz), beta2 (19–21 Hz), and beta3 (22–30 Hz). Additionally, we adjusted the alpha and theta bands based on each individual’s alpha peak frequency (IAF). The results revealed a significant decrease in the alpha frequency compared to the controls. The WAIS-total and MMSE showed a positive correlation with the alpha frequency in DS. Moreover, negative correlations were observed between WAIS and RBM scores and occipital alpha power. When examining absolute power in fixed bands, DS exhibited significantly higher absolute power and increased CSD in theta, alpha1, and beta1 compared to controls. WAIS was negatively correlated with CSDs in the right frontal lobe and right PCC, while RBM showed a negative correlation with CSD in the right BA9. A negative correlation was identified between absolute occipital power and cognitive test performance using 1 Hz bands. When analysing absolute power in IAF-adjusted bands, the control group exhibited maximum alpha power in the upper alpha band, while individuals with DS showed it in the lower alpha2 band. DS demonstrated a significantly increased CDS in all individually adjusted bands compared to controls. In the theta band, WAIS exhibited a negative correlation with CSD in the right BA9. DS, when compared to controls, displayed significantly higher absolute power in the theta and alpha bands. Regarding relative power, the maximum value of alpha was found in alpha1 in DS and alpha2 in controls. A notable difference between groups was observed in alpha2, indicating a decreased CSD in the cingulate cortex in DS. However, no correlations were found between the regions of decreased CSD in alpha2 and cognitive scores. The average value for relative alpha2 and its occipital values correlated with cognitive scores, showing a negative correlation with power at 7–8 Hz and a positive correlation at 11–12 Hz.

In the research study [30], EEG recordings were collected from 36 adults with DS aged 16–56 years. The study identified the presence of significant cognitive decline by assessing data from the Cambridge examination of mental disorders of older people with Down syndrome and others with intellectual disabilities (CAMDEX-DS). All participants were expected to exhibit no decline based on the responses to this questionnaire. All participants had genetically confirmed trisomy 21. General cognitive ability was estimated using Kaufmann’s brief intelligence test, second edition (KBIT-2) raw test scores. Participants kept their eyes closed during the recordings. The study generated mean scalp maps for theta (5 Hz) and alpha (8 Hz) power to examine their regional distribution and identify the location of maximum power within each band. Linear regression was employed to explore the relationship between the raw KBIT-2 scores and both the alpha peak amplitude (i.e., maximum power within the 8–13 Hz range) and the APF (i.e., the frequency within the 8–13 Hz range where peak amplitude occurs) in the spectra derived from regional electrode averages. These regions included occipital electrodes (E70, E71, E74, E75, E76, E82, E83) and frontal electrodes (E4, E5, E10, E11, E12, E16, E18, E19). Additionally, a scalp-wide statistical parametric mapping (SPM) of power in the combined theta–alpha (4–13 Hz) range was generated. Regression analysis using a general linear model aimed to identify significant associations between the raw KBIT-2 scores and theta–alpha power across the scalp. The findings revealed that average scalp maps and mean power spectra indicated a frontal dominance of theta activity and a posterior distribution of alpha activity for all participants. Linear regression demonstrated a significant association between the raw KBIT-2 scores and the peak alpha amplitude at the frontal region. However, the relationship between the peak alpha amplitude at the occipital region and the raw KBIT-2 scores did not reach significance. No significant association was found between the peak alpha frequency and the KBIT-2 scores for frontal or occipital electrodes. The whole-scalp SPM analysis of spectral estimates uncovered clusters of significant positive correlations between estimated general cognitive ability (raw KBIT-2 scores) and power in the 4–13 Hz range.

## 4. Discussion

Research conducted on EEG in individuals with DS has shown distinct and regular patterns of brain activity compared to typically developing people. EEG recordings of individuals with DS have revealed elevated slow-wave activity, especially in the delta and theta frequency bands; these findings have been found to be consistent with most research. Furthermore, anomalies have been discovered in the alpha and beta frequency bands in DS patients. Again, multiple studies consistently indicate that individuals with DS exhibit lower amplitude and power of alpha activity, which is linked to relaxed wakefulness [31] and visual attention [32], especially in the posterior brain regions [31]. Conversely, beta activity, related to cognitive processes such as attention and working memory [33], displays inconsistent findings in DS patients, with some studies reporting increased amplitude while others suggesting reduced beta activity. These observations suggest disruptions in neural oscillations and functional connectivity in DS.

Interestingly, several studies have explored the correlations between EEG patterns and cognitive function in DS individuals. Some researchers have reported associations between slower alpha rhythms and cognitive deficits, indicating that alterations in alpha activity might be linked to impaired cognitive performance in DS. Conversely, other studies have shown that a higher alpha EC/EO ratio, reflecting alpha activity during eye closure and eye opening, aligns with better cognitive performance in DS patients. This correlation highlights the potential utility of EEG measures in assessing cognitive abilities and monitoring cognitive changes over time in DS individuals. One study investigated the relationship between EEG alpha activity and visuospatial skills in older DS patients. The results showed that DS patients with an abnormal EEG background had a significant impairment in visuospatial skills compared to those with a normal alpha background. This suggests that EEG measures could serve as potential markers for specific cognitive deficits in DS, aiding in early detection and targeted interventions. This may be made possible by the increasing use of portable EEG measurement devices at home. Knowing the frequency bands with deviations in activity encourages testing the effectiveness of neurofeedback, a method for consciously influencing EEG activity [34]. The study by Surmeli and Ertem [35] showed that neurofeedback training, inhibiting theta and alpha or increasing alpha, significantly improved the results on WISC-R and TOVA.

Furthermore, longitudinal studies have provided insights into the evolution of EEG patterns with age in DS patients. With advancing age, individuals with DS commonly exhibit a decline in alpha activity, particularly noticeable in posterior brain regions. This age-related decline in alpha activity might correlate with cognitive decline and neuropathological changes characteristic of aging in DS. These findings emphasise the importance of understanding the neural dynamics across the lifespan of individuals with DS to provide targeted interventions for cognitive support and enhancement Figure 2 shows changes in EEG alpha, theta, and delta activity across different age groups in people with DS.

Studies exploring EEG coherence, reflecting the synchronisation between different brain regions, have uncovered altered patterns in DS. Altered EEG coherence patterns have been observed, indicating disruptions in functional connectivity within the brain among individuals with DS. Such connectivity disruptions may contribute to the cognitive impairments seen in DS patients, shedding light on the neural basis of their cognitive profile.

Apart from cognitive function, some studies have explored the relationship between EEG patterns and age-related neuropathological changes in DS individuals. Notably, older DS patients with loss of alpha background activity on EEG were found to have dementia, with some cases confirmed to have neuropathological signs of Alzheimer’s disease. This suggests that EEG measures might have potential as biomarkers for identifying individuals at risk of developing dementia in DS, allowing for early intervention strategies and personalised medicine. By detecting declines in alpha activity and associated cognitive decline early enough, we can implement medications that target the declines in cognition and alpha activity (discussed later).

Only one study showed results in the gamma band (reduced power density). This type of band is relatively little known and is strongly influenced by muscle artifacts. For this reason, measuring this band in people with DS may be difficult due to frequent motor hyperactivity [36].

Studies have shown mixed results when it comes to beta activity. Some studies have shown increased beta activity in people with DS, and some studies have had the opposite results. Several factors may have contributed to this, such as beta-amyloid load (amyloid-positive patients with AD show a decrease in [37] or an increase in beta power [38]). It is not known how beta power relates to amyloid load in the context of DS. Taking benzodiazepines causes an increase in beta activity [39], and some DS patients have taken these drugs. Furthermore, beta activity is increased under stress [40], and people with DS often experience anxiety disorders [41]. To address all of these inconsistencies, it is crucial for future studies to control for all factors that may influence beta activity in people with DS.

Most studies measured EEG with eyes closed. In this condition, alpha activity is dominant, especially in the occipital area [42], so it is easier to measure. However, in several studies, patients had their EEG measured while their eyes were open. This makes comparisons between studies that differ in these conditions difficult.

Overall, the research on EEG in DS has unveiled a complex interplay between neural activity, cognitive function, and age-related changes. The consistent findings of altered EEG patterns and their associations with cognitive deficits offer potential avenues for developing novel therapeutic interventions to address specific cognitive impairments in DS. Moreover, the use of EEG measures as biomarkers for identifying individuals at risk of dementia and monitoring disease progression could have significant clinical implications.

### 4.1. Interpretation of the Results of Consistent Findings Regarding Altered EEG Activity in Individuals with DS

#### 4.1.1. Alpha Activity

Researchers have observed diminished levels of alpha brainwave activity during rest in various neurodegenerative disorders and DS, potentially linked to issues in cholinergic function [43]. Initially, alpha waves were believed to indicate a lack of brain activity or relaxation due to their inverse connection with neuronal activity [44,45,46]. However, our current understanding reveals that alpha waves actively inhibit brain activity, help with selective attention, and support executive control processes [47,48]. These alpha oscillations come from interactions between the thalamus and cortex, as well as within the cortex itself [49,50,51], and they are modulated by the neurotransmitter acetylcholine (ACh) [52,53,54]. The cholinergic hypothesis proposes that reduced release of acetylcholine might be a cause of symptoms in DS [43]. The alpha rhythm seen in EEG recordings is affected by cholinergic signalling and relates to thalamic alpha oscillations [53]. Changes in cholinergic signalling can alter the thalamic alpha rhythm in DS, leading to a decrease in both alpha power and frequency. Interestingly, medications that impact cholinergic pathways and are used to treat dementia have been found to affect brainwave patterns in people with Alzheimer’s and Parkinson’s disease-related dementia. These medications increase resting alpha and beta wave power while decreasing theta and delta wave power [55,56,57]. Although people with DS are generally excluded from research on antidementia drugs [58], several studies have demonstrated their effectiveness. For example, a research study by Eady et al. [59] following 310 individuals with DS and Alzheimer’s disease found that those who received cholinesterase inhibitor treatment showed similar positive results in terms of cognition and behaviour compared to individuals with sporadic AD. Notably, individuals with DS who received either a single cholinesterase inhibitor or a combination of them had a median survival rate of approximately 5.6 years after diagnosis. This was an improvement compared to those who did not take any medication, as their median survival rate was around 3.4 years.

#### 4.1.2. Delta Activity

Delta activity, which is characterised by slow-wave oscillations around 2 Hz, is regulated by the thalamocortical (TC) system, specifically involving the thalamic reticular nucleus (TRN) and TC relay neurons [60,61]. In the brain, GABA (gamma-aminobutyric acid) is a crucial inhibitory neurotransmitter that plays a significant role in controlling neuronal excitability [62]. The generation of delta waves, associated with delta frequency burst firing, is influenced by GABAergic inhibitory interneurons and their interaction with GABAA receptors [61]. The heightened delta activity observed in individuals with DS can be connected to GABA mechanisms in several ways. Firstly, GABA’s role in modulating neuronal excitability and controlling delta wave generation is crucial. Delta frequency burst firing, responsible for delta wave activity, can be triggered and enhanced by inhibitory inputs through GABAA receptors, particularly those composed of α4βδ subunits, leading to delta wave production [63]. This suggests that GABA’s inhibitory effects on TC relay neurons contribute to the excessive delta activity seen in DS. Secondly, the GABAergic system’s role in DS involves complex dynamics. While GABA levels may not significantly differ in certain brain regions [64], alterations in GABAergic interneurons have been observed [65]. An increase in the number of inhibitory interneurons, particularly those expressing calretinin (CR) [66] and somatostatin (SST) [66], may contribute to heightened inhibitory activity and potentially excessive delta activity in DS. Lastly, the composition of GABAA receptors, which mediate GABA’s inhibitory effects, plays a role. Different subunit combinations of GABAA receptors determine their properties and locations. Studies have reported altered expression of specific GABAA receptor subunits in DS, such as reduced levels of α5 and γ2 subunits in the hippocampus [62]. Changes in GABAA receptor composition and distribution, including those containing α1, α2, α3, and δ subunits, could disrupt the balance between excitatory and inhibitory signalling, potentially contributing to excessive delta activity.

#### 4.1.3. Theta Activity

Theta oscillations arise from coordinated neuron activity in specific brain regions, notably the hippocampus [67,68,69]. In individuals with DS, modified neural connections might influence the generation and synchronisation of theta oscillations. The hippocampus, crucial for memory formation and spatial navigation [70,71], significantly contributes to these oscillations [72]. Structural and functional irregularities in the hippocampus are common in those with DS [73], potentially affecting the usual generation and control of theta oscillations. Theta oscillations are linked to various cognitive processes like attention, memory, and learning [74], all of which are often compromised in people with DS [75] Changes in theta oscillations might mirror the cognitive challenges in this group. DS can disrupt neurotransmitter systems like the cholinergic system [43], which plays a role in theta oscillations [76]. Altered cholinergic activity might disrupt the delicate equilibrium required for theta oscillations, potentially causing excessive theta rhythms. The last possible explanation for increased theta in DS might involve alterations in the medial septum–diagonal band of Broca (MS-DbB), which is pivotal for theta generation. This structure influences hippocampal activity and theta oscillations. Studies indicate that the rhythmic activity of MS-DbB neurons corresponds with theta waves in the hippocampus [67]. The unique genetic makeup of individuals with DS could lead to an abnormal functioning or connections within the MS-DbB circuit, potentially causing an imbalance in theta generation [67].

### 4.2. Sources of Inconsistencies in Results and Prospects for Avoiding Them in Future Research

While studies have identified three consistent EEG patterns in DS, discrepancies in results persist, and it is essential to highlight potential sources of these inconsistencies. First, the research was conducted over several decades, witnessing advancements in EEG technology, leading to increased diagnostic sensitivity and accuracy. Second, variations in the definition of frequency bands, subbands, and the choice between relative and absolute power measurements across studies hinder direct comparisons. Future research should consider adopting the frequency bands used in prior trials, potentially exploring subbands for nuanced insights. Third, diverse lead placements for EEG activity measurement were employed. We recommend that future studies adhere to the international 10–20 system for electrode mounting, utilising at least 21 leads [77]. Fourth, various artifact correction methods were used—from the most basic handmade methods to computerised technologies. We propose that future studies combine both artifact correction techniques. The following articles, [78,79], can help you find the appropriate methods. Fifth, various methods were used to statistically analyse the results. Sixth, various EEG analysis methods were used. For detailed information on EEG signal analysis methods, please refer to the reviews [80,81]. Seventh, variations in patient demographics, particularly age, were observed, impacting EEG results. Some studies did not divide patients into, for example, younger adults and older adults, and created a general group with all patients. EEG changes with age and aging [82,83], so these changes should be taken into account when analysing the results. Additionally, differences in amyloid burden between children and adults with DS should be taken into account. Eighth, not all studies excluded comorbid neurological and psychiatric diseases that could influence EEG results. Therefore, future studies should comprehensively examine patients with DS and exclude people who have neurological and psychiatric disorders. Ninth, not all studies excluded patients with epilepsy. Epilepsy is common in DS [84] and affects the EEG spectrum [85]. It is necessary to control this neurological condition and exclude patients with epilepsy in future studies or include only patients with epilepsy, keeping in mind the subtypes of epilepsy. Tenth, cognitive decline and Alzheimer’s disease presence were inconsistently assessed, despite being common in DS. Future studies should test patients for AD and either exclude patients with AD from the study or create a subgroup with only AD patients. To test for cognitive decline, we suggest using the Cambridge examination of mental disorders of older people with Down syndrome and others with intellectual disabilities (CAMDEX-DS), which is adapted to people with DS [86]. Eleventh, the studies used different cognitive measures, which also makes comparisons between results difficult. DS, as a disorder, is characterised by differences in developmental outcomes, such as intellectual disability, language, memory, executive functions, adaptive behaviour, maladaptive behaviour, and emotional functioning [87], and these differences should be taken into account by creating separate groups in future studies, based on cognitive results. Kaufmann’s brief intelligence test, second edition (KBIT-2) [88] can be used to measure general cognitive abilities. Twelfth, the included studies examined EEG with eyes open or closed (predominantly the latter). We suggest that future studies examine EEG in both conditions or with eyes closed due to the predominant alpha activity, which is most easily measurable in the eyes-closed condition and is an important cognitive marker in DS. Additionally, general recommendations for future research include the use of large patient samples, the inclusion of an age-matched healthy control group, and EEG absolute and relative power measurements. To extend these results, future studies could add neuroimaging diagnostics such as fMRI to EEG diagnostics to deepen our understanding of DS and uncover common neural mechanisms.

## 5. Conclusions

Ultimately, EEG studies in individuals with DS offer crucial insights into the neurological origins of cognitive challenges in this population. The repeated observations of changed EEG patterns and their correlations with cognitive impairments demonstrate the potential value of EEG measures as biomarkers for tracking progression of the disorder and cognitive assessments. These abnormal EEG patterns, especially in slow-wave and alpha activity, point to a disruption in neuronal synchronisation and communication in DS, which may be one reason why people with this condition face cognitive difficulties. Moreover, connections between cognitive function and EEG parameters highlight the potential value of EEG as a non-invasive, objective method of evaluating cognitive performance in people with DS.

Longitudinal studies have been instrumental in unveiling the neurodevelopmental trajectory of DS and understanding how EEG patterns change with age. Comprehending the neurobiological alterations that occur in DS patients throughout the course of their lives is essential for creating targeted interventions that enhance cognitive performance and overall quality of life. Age-related variations in EEG were observed in these longitudinal studies, suggesting that the brain activity in DS changes with time and may be a reflection of the progressive nature of cognitive impairments in this population. This insight is pivotal for tailoring interventions that address the diverse needs of individuals across different life stages. Despite difficulties with heterogeneity and limited sample numbers, this area of study has much promise for expanding our knowledge of the neurobiology and cognitive deficits associated with DS. Further exploration of EEG measures and their correlation with cognitive function will continue to deepen our comprehension of the intricacies of DS neurobiology.

## Figures and Tables

**Figure 1 brainsci-14-00136-f001:**
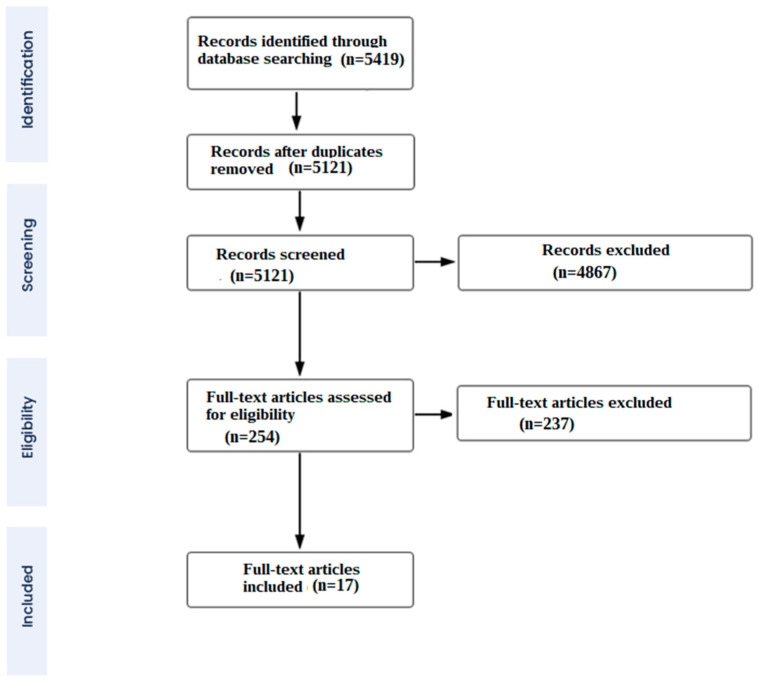
Flow chart depicting the different phases of the systematic review.

**Figure 2 brainsci-14-00136-f002:**
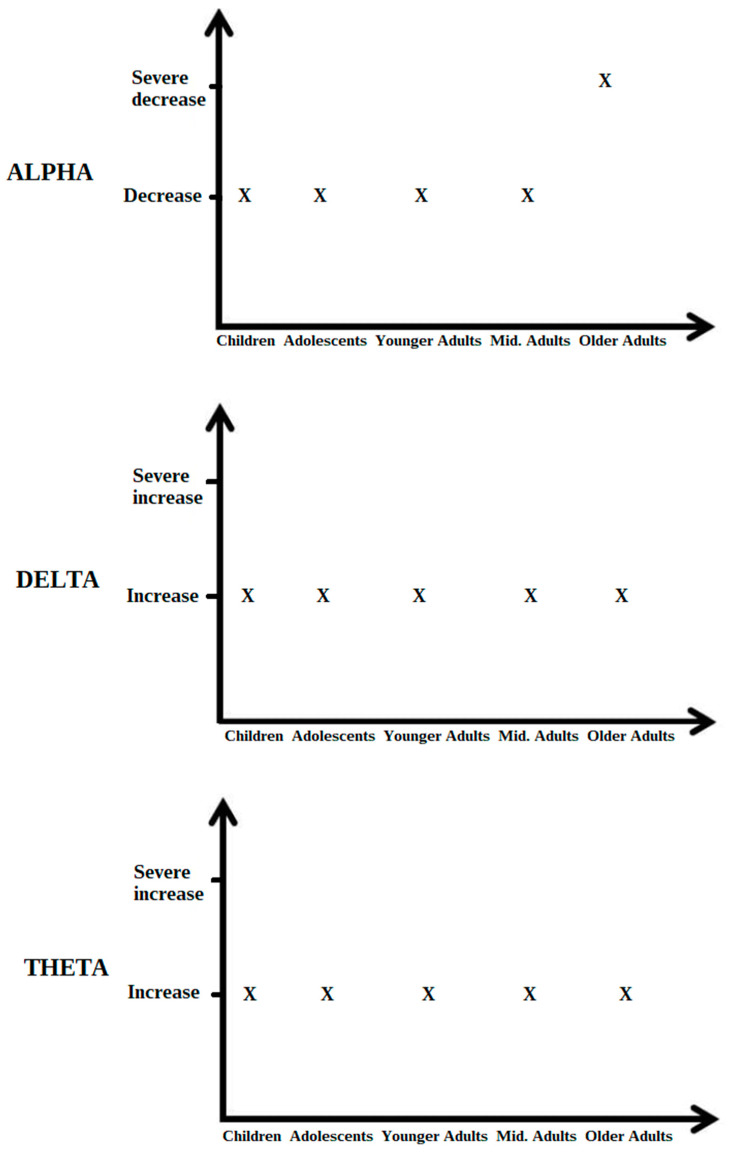
Changes in EEG alpha, theta and delta activity over age in people with DS.

**Table 1 brainsci-14-00136-t001:** Studies included in the review.

Study	Participants	Methods of EEG Analysis	Controls for Confounders	Findings
[14]	- 19 young adults with DS, aged 19 to 37 years, and 9 older adults, aged 42 to 66 years; - 13 healthy control subjects aged 22 to 38 years.	A certified electroencephalographer analysed the EEGs, categorising them based on the presence or absence of alpha background activity.	+ DS was confirmed by genetic tests; + Presence of a control group; + Patients divided into two groups; + The patients did not have epilepsy; + Patients had no other comorbidities; + Patients were not taking psychotropic medications; - Four elderly patients had a history of mental disorientation, deterioration, and memory loss and suffered from dementia, but it was not stated which diagnostic tests were used for this purpose.	In the group of 9 older DS patients, 5 (55.6%) had normal alpha background activity, and 4 (44.4%) had abnormal activity; Some of the older DS patients with abnormal EEG readings exhibited specific neurological patterns, including frontal slowing or sharp transients in certain brain regions; All older DS patients with a loss of alpha background in their EEGs showed signs of dementia, with two confirmed cases of Alzheimer’s disease upon postmortem examination; In young adult DS patients, no significant differences in neuropsychological findings were observed between those with normal and slower EEG backgrounds; Among older DS patients, those with abnormal EEG backgrounds demonstrated notable impairments in visuospatial skills and attention compared to those with normal alpha background; Older DS patients showed a non-significant decline of at least 50% in language function, one attention test, and the test of visual recognition memory compared to older patients with normal alpha background activity.
[15]	- 32 adults with DS (age 21–60 years); - 31 age-matched controls (age 21–60 years).	The absolute amplitude and power for standard EEG bands, along with peak frequency and mean frequency were calculated. In terms of reactivity, EC/EO ratios for amplitude, power, and the percentage diminution of amplitude and power across all bands were computed. A clinical neuropsychologist evaluated the EEG tracings without knowledge of the patient’s details, grading them for different abnormalities, such as the slowing of the dominant occipital rhythm.	+ Presence of a control group; - There was no information whether DS was confirmed by genetic tests; - There was a wide range of variability in mental deficiency, with standard deviation values being significantly larger than the means. It was not reported how many patients had experienced significant cognitive decline or confirmed dementia; - Patients were taking neuroleptics and benzodiazepines; - Patients with comorbidities were not excluded; - Some patients had epilepsy.	Analysis of conventional EEG readings revealed that among the patients, 16 had a normal background rhythm in the occipital region, 12 displayed mild slowing, 3 exhibited moderate slowing, and only 1 patient showed marked slowing; When comparing DS patients to controls, the study found elevated levels of delta, theta, alpha, and beta activity in the DS group; Specifically in the alpha band, researchers noted a significant reduction in the amplitude ratio between EC and EO in DS patients compared to the control group; The study identified meaningful connections between various neuropsychological measures, the MMS, the Blessed scale, and the pooled alpha EC/EO ratio; A positive correlation indicated that a higher alpha EC/EO ratio was associated with better cognitive performance; The alpha EC/EO ratio demonstrated an inverse correlation with the age of the patients, suggesting that older patients tended to have a lower ratio.
[16]	- 38 adolescents with DS (18.7 years ± 0.67 SE); - 17 age-matched healthy controls.	The spectral analysis was conducted using the FFT with the Welch technique, and interhemispheric directional EEG functional coupling was analysed using the Directed Transfer Function (DFT). The standard frequency bands examined included alpha1 (8–10.5 Hz), alpha2 (10.5–13 Hz), beta1, and beta2. Statistical analyses were performed on subgroups, bands and electrodes.	+ DS was confirmed by genetic tests; + People with DS were of similar age; + Presence of a control group; + Patients with comorbidities were excluded; + Epilepsy was excluded in the patients; + Patients were not taking psychoactive drugs.	Power density values in the DS group in the delta frequency band noticeably increased across the frontal areas of the scalp; The DS group showed reduced power density values in the alpha, beta, and gamma frequency bands.
[17]	- 45 young adults with DS (mean age 22.8 years); - 45 age-matched healthy adults (mean age 22.4 years).	The analysis involved the examination of both Alpha 1 (8–10.5 Hz) and Alpha 2 (10.5–13 Hz) bands. Cortical EEG sources were estimated using LORETA, wherein relative power current source densities at cortical voxels were compared to address abnormalities caused by head volume conduction. The Alpha frequency was calculated over an extended range.	+ DS was confirmed by genetic tests; + All individuals recruited for this study were confirmed to be free from dementia; + Participants were of similar age; + Presence of a control group; + Patients with comorbidities were excluded; + Epilepsy was excluded in the patients; + Patients were not taking psychoactive drugs.	There was a reduction in the amplitude of sources within the alpha 1 and alpha 2 frequency bands, accompanied by an increase in delta band activity; Further analysis, conducted after the initial findings, revealed that the pattern of sources in the control group was higher than that in the DS group. This difference was evident in the LORETA solutions, highlighting specific brain regions; Heightened activity alpha 1 and alpha 2 sources in central, parietal, occipital, temporal, and limbic regions the control group. Similarly, in beta 1 sources, heightened activity was observed in parietal, occipital, temporal, and limbic areas in the control group; An opposing trend was noted in occipital delta sources, where the DS group exhibited different patterns compared to the control group.
[18]	- 88 children with DS (aged six months, up to five years); - 277 age-matched healthy children.	A time code was automatically generated and recorded, enabling the automated identification of EEG periods free from artifacts that were visually selected on analog tape for analog-to-digital conversion. To conduct additional analysis, the spectra of the two fronto-central and the two parieto-occipital leads were segmented into the six frequency bands.	+ People with DS were of similar age; + Presence of a control group; - There was no information whether DS was confirmed by genetic tests; - No information about the occurrence of epilepsy in patients; - No information on the occurrence of comorbidities in patients; - No information about medications being used.	When examining EEG development in the fronto-central region among younger children with DS, minor differences in subdelta, delta, and theta bands were noted compared to their typically developing peers; As these children aged, there was a gradual increase in theta activity, peaking around two years of age; The most notable deviations from normal EEG development occurred in the alpha and beta 1 bands; In the first year of life, average alpha band values in DS EEGs were within one standard deviation of the control group; Between ages two to five, a distinct reduction in alpha activity was evident, particularly noticeable at ages two and three; In the parieto-occipital area, a gradual increase in relative subdelta, delta, and theta activity occurred with age, marked by a rise in theta activity at two years and increased subdelta and delta activity at five years; Similar deviations in the alpha and beta 1 bands were observed in the parieto-occipital region, becoming more pronounced as children with DS grew older; A similar pattern emerged when comparing the frontal and posterior regions, with a more prominent reduction in relative alpha and beta power as age increased; While the development of low frequencies was evident, older children with DS showed a relative increase in these low frequencies; Regarding the absolute power of different frequency bands, several findings emerged. Over the fronto-central area, subdelta, delta, theta, and beta 2 power were higher in DS children, while absolute alpha power showed a deficit; In the parieto-occipital area, the absolute power of different frequency ranges was higher in DS children, particularly due to increased subdelta, delta, and theta power; In some instances, absolute alpha power in DS children was higher, up to the age of three years, but reduced in four- and five-year-olds. Higher frequency bands like beta 1 and beta 2 also showed a significant increase in absolute power.
[19]	- 40 people with DS (age 15–54, mean 30.3 ± 11.4 years); - 42 non-DS with intellectual disabilities (mean age 30.9 ± 10.8 years); - 47 healthy control group (mean age 30.2 ± 11.5 years).	The analysis involved examining relative power using EEGs from four derivations in the left hemisphere (O1-A1, C3-A1, P3-A1, F3-A1). Specifically, five artifact-free epochs of 5.12 seconds recorded from O1, were subjected to FFT. The FFT was employed to analyse the peak frequency of occipital alpha rhythms, defined as the frequency with the most prominent peak in the 8–13 Hz band. Relative power in six bands was then calculated in the 4–30 Hz range (theta1, theta2, alpha1, alpha2, beta1, beta2).	+ Presence of control groups; + Patients with epilepsy were excluded; + Patients with neurological diseases were excluded; + Patients did not use any medications; - There was no information whether DS was confirmed by genetic tests; - Wide age range in all study groups; - It was not stated whether the patients had symptoms of dementia.	A significant inverse relationship between the peak frequencies of alpha rhythms in the left occipital location (O1-A1) and the chronological age of individuals in the DS group; Comparison with a control group revealed several age-related differences in relative power; In the youngest age group of DS individuals, there were noteworthy increases in the relative power of the theta1 and beta2 bands across all locations, along with an increase in the beta1 band in the F3 and C3 locations; The 25–34 age group also exhibited increased relative power in the beta1 and beta2 bands; Among the DS 35–44 age group, significant increases in the relative power of the theta2 and alpha1 bands were observed; Individuals aged 45–54 showed increased relative power in the theta2 and alpha1 bands, particularly in the occipital EEG; Across all age groups of DS, the relative power in the alpha2 band consistently remained lower compared to the control group, especially in the posterior leads.
[20]	- 31 people with DS (mean age 35 ± 10 years); - 69 AD patients (mean age 80.1 ± 9.5 years); - 16 elderly controls (age 81.6 ± 7.1 years); - 26 young healthy controls (age 26 ± 8 years).	Absolute amplitude and power of typical EEG bands (alpha 7.57–13.92 Hz), along with peak frequency and mean frequency (for both the entire spectrum and a combination of alpha and theta = 4.15–13.92 Hz), were calculated. A clinical neurophysiologist assessed the EEG tracings without knowledge of the patient’s details, grading them for various abnormalities, including the slowing of the dominant occipital rhythm.	+ The DS patients were divided into two groups: those under 40 years old (17 individuals) and those aged 40 and older (10 individuals); - There was no information whether DS was confirmed by genetic tests; - The individuals with DS exhibited cognitive impairment, as assessed through MMS scores, but there was no information that they have been diagnosed with dementia; - Patients from the DS and AD groups were taking neuroleptics and benzodiazepines; - Six AD patients and five DS patients had a history of epileptic seizures; - No information on exclusion of patients with comorbidities.	The findings revealed significantly elevated levels of slow wave activity, especially in the theta and delta bands among DS patients; Older patients exhibited lower peak frequency (8.3 ± 0.8 Hz vs. 9.0 ± 0.7 Hz) and mean frequency (8.6 ± 0.5 vs. 9.0 ± 0.4 Hz) within the 4.15–13.92 Hz range compared to the younger group; Age consistently correlated with frequency parameters across the entire DS group and specifically in DS patients aged 40 and above; Within the DS group, peak frequency showed a correlation with cognitive scores (measured by the MMS score) and functions related to vision, motor skills (praxic), speech, and list learning; Significant correlations were also observed between delta amplitude and praxic functions, as well as list learning in DS patients.
[21]	- 32 people with DS (age 20–46 years); - 15 healthy older adult controls (in their 60s); - 15 healthy younger adult controls.	The quantity of waves present in 10 frequency bands (delta, theta1–3, alpha1–4, beta1–2) was determined using the “wave-form recognition method.” Additionally, the mean frequency of the occipital region (O1) was computed within the theta-alpha band range of 4–13 Hz.	+ It was not reported that any of the patients had been diagnosed with dementia, but test results indicated cognitive decline; + Patients with DS were divided into three groups; + One patient was taking anticonvulsants; - 90% of patients had genetically confirmed trisomy 21, and 10% of patients had the mosaic type; - Epilepsy was not excluded in the patients; - Comorbidities in patients were not excluded.	Comparisons among the three age groups of DS individuals uncovered heightened theta 1 activity at O1 and increased theta 3 activity at seven sites, mainly in the frontal and parietal regions, in individuals in their 40s compared to those in their 20s; Alpha 4 and beta 1 activities generally decreased across 11 and 8 sites, respectively. In closer examination, comparing the 30s group to the 20s group revealed a higher incidence of delta activity at one site (Fp2) and a reduction in alpha 3 activity at one site (F3). Alpha 4 activity was diminished at six sites, mainly in frontal and parietal regions; When comparing the 40s and 30s groups, a few sporadic differences were observed in specific sites and frequency bands. Specifically, theta 1 activity was more frequent at one site (O1), theta 3 was more frequent at two sites (F4, P4), and alpha 4 activity was less frequent at one site (P4); A decreasing trend in mean frequency with advancing age within the DS group, with values of 9.37 Hz for the 20s group, 9.17 Hz for the 30s group, and 8.76 Hz for the 40s group. A notable difference was found between the 20s and 40s age groups.
[22]	(1) Cross-sectional study: - 265 children and adults with DS (age range 8–55 years) - 242 non-DS children and adults with intellectual disabilities (age range 7–58 years); - 239 healthy children and adults (age range 2–59 years); (2) Longitudinal study with period between 8–9 years: - 28 people with DS; - 14 non-DS with intellectual disabilities.	The dominant frequency was determined based on the peak of the average spectrum. The analysis specifically targeted the frontal, central, and occipital regions.	+ The results were divided into age groups; + Epilepsy was not reported in the patients; - For one-third of the individuals with DS, a chromosome karyotype was not conducted. Among the remaining two-thirds, the karyotype was typical trisomy, except for four individuals who exhibited a mosaic type; - No information on exclusion of patients with comorbidities; - No information about medications used by patients.	In the occipital brain region, the prevalence of the 9 Hz frequency and the 10 ± 12 Hz range in individuals aged 10 ± 14 years and 15 ± 19 years did not significantly differ from the patterns observed in the healthy control group; For individuals under 9 years old, the predominant component was identified within the 8 and 9 Hz range, with a comparatively lesser presence in the 10 ± 12 Hz range; Between the ages of 10 ± 14 years, the dominance of the theta frequency band diminished, and more than half of the subjects in this age group began to show dominance at 10 ± 12 Hz; In contrast, the 9 Hz component emerged as the most dominant during the subsequent 20 ± 24 years, continuing until the 35 ± 39 year range. An increasing dominance of the 8 Hz component was also observed during these time periods, reaching nearly the same prevalence as the 9 Hz component in the 30 ± 34 and 35–39 year ranges; In the subsequent 40 ± 44 years, the prevalence of the 8 Hz component surpassed that of the 9 Hz component, becoming the most prevalent after the age of 45; For the central brain area, the prevalence of the 4 ± 5 Hz range was unusually high in the 20 ± 24 year range, raising concerns about potential artifacts. Therefore, the analysis focused on higher frequency ranges excluding 4 ± 5 Hz; The prevalence of 9 Hz was similar to that of 10 ± 12 Hz for individuals aged 10 ± 14 years, and the 9 Hz component became slightly more prevalent in the 15 ± 19 year range. It reached its peak prevalence in the 25 ± 29 year range but began to decrease thereafter; The prevalence of the 8 Hz component exhibited a gradual increase, surpassing other frequencies in prevalence from the age of 30 ± 34 years onward; Similar trends were observed in the frontal EEG, with the prevalence of 9 Hz notably high in the youngest group but generally consistent with the central EEG in later ages; Among those with DS, the dominant frequency component was 9 Hz or higher for all subjects in their twenties. However, in their thirties, an increase in individuals showing frequency components lower than 9 Hz was observed (10 out of 20, 50.0%), and this trend continued in their forties (15 out of 19, 78.9%); Among subjects in their fifties, three out of four subjects examined (75.0%) displayed this lower frequency pattern; In terms of the age at which the peak frequency in spectra decreased in the follow-up subjects, distinct drops were observed in the thirties for seven individuals and for two individuals in the forties. However, a few individuals maintained a dominant frequency of 9 Hz or higher even beyond the age of 45.
[23]	- 25 adults with DS (age range 17–44); - 25 healthy controls (age range 16–44).	Absolute and relative power measures for each frequency band of interest (delta 0.5–4 Hz; theta 4–8 Hz; alpha 8–13 Hz; beta 13–30 Hz) were obtained for each region (frontal and occipital) for every individual. Additionally, alpha peak features were calculated, with the frequency of the peak amplitude within the 8–13 Hz range being defined. Some individuals with DS did not exhibit a detectable alpha peak. To address this, alpha peak features were derived for all participants by eliminating the linear trend from individual power spectra, a process referred to as ‘spectral normalisation’.	+ Every participant had their trisomy 21 genetically confirmed; + To be eligible for the study, all participants had to exhibit no decline according to this questionnaire; + Age-matched participants in control group; + Participants with an acute physical or mental health condition were excluded; - The presence of epilepsy in patients was not excluded; - No information about medications used by patients.	When scrutinising EEG data from the occipital region, significant disparities emerged between individuals with DS and those with typical development controls; Those with DS displayed considerably higher absolute and relative power in the delta band, coupled with relatively higher power in the theta band. Conversely, there were marked reductions in both absolute and relative power within the alpha and beta bands. Additionally, the alpha peak amplitude was notably lower in the DS group, with the most substantial differences found in relative alpha power, accounting for 56.5% of the variation between the two groups; Similar patterns were observed in the frontal region, where all group differences, encompassing both absolute and relative values, were statistically significant, except for absolute alpha and beta power and alpha peak frequency; In the frontal region, individuals with DS demonstrated notably elevated absolute and relative power values in the delta and theta bands. In contrast, relative power values in the alpha and beta bands, as well as both absolute and relative alpha peak amplitudes, were considerably lower. The most pronounced differences were observed in absolute alpha peak amplitude and relative alpha power, accounting for 28.8% and 20.9% of the group variation, respectively.
[24]	- 13 adults with DS (mean age 33.8 years); - 13 healthy age-matched controls.	Spectral analysis was conducted using the Dirichlet window and FFT. The O2-A2 electrodes were utilized for analysis, with alpha defined within the range of 8–12.9 Hz. The dominant occipital frequency was determined as the frequency with the largest power peak in the 2–20 Hz range in the T6-O2 channels.	+ All participants had the classical phenotype of DS; + All included patients were without dementia; + Presence of an age-matched control group; + Patients with focal neurological findings were excluded; + Patients were not taking psychotropic medications.	Individuals with DS displayed elevated absolute power in the delta, theta, and beta bands, along with overall power in the resting EEG; When comparing the absolute and relative powers within specific frequency ranges between DS subjects and control subjects, those with DS exhibited significantly higher values in the 4.5-Hz and 8.8-Hz frequency ranges; Concerning cognitive abnormalities, noteworthy negative correlations were observed between resting power in the 4.5-Hz and 8.8-Hz frequency ranges and cognitive test performance.
[25]	- 6 people with DS (24–78 years, mean age 42.3 years).	Absolute power, relative power, symmetry, and phase coherence were measured. Age-appropriate normative values were used for statistical comparisons of the variables. Additional EEG analyses, such as multivariate analyses of composite features across frequency bands and electrode locations, were also conducted.	- Significant age range of the participants; - No information about cognitive decline or dementia in participants; - No control group; * Lack of access to information about the genetic confirmation of DS, the presence of coexisting neurological and psychiatric diseases, epileptic seizures and psychoactive drugs taken.	The results showed an elevation in both absolute and relative power within the delta and theta frequency bands; This increase was concurrent with a simultaneous reduction in alpha and beta activity.
[26]	- 36 people with DS (aged 15–54, mean age 30.7 ± 11.5 years); - 47 healthy controls (mean age 30.9 ± 10.8 years); - 42 non-DS mentally retarded (mean age 30.9 ± 10.8 years).	Measured the peak frequency and relative powers within conventional EEG bands, with a specific division for the alpha band into the ranges of 8–10.5 Hz and 10.5–13 Hz.	+ DS was confirmed by genetic tests; + Presence of a control group; + To explore age-related changes, the subjects were categorised into four age groups: (1) 15–24, (2) 25–34, (3) 35–44, and (4) 45–54 years old; + Patients with DS did not take any psychoactive drugs; - No information about cognitive decline or dementia; - No information about coexisting neurological and psychiatric diseases.	The study identified a substantial negative correlation between alpha peak frequency and age, demonstrating a linear pattern, a trend not observed in both control groups; Significant disparities in alpha frequency were observed across all age groups in individuals with DS, even within the 15–24 age range; In contrast to healthy controls, individuals with DS showed no variations in beta and a noteworthy reduction in relative power in alpha2 for ages 25–54; Older adults (35–54 years) displayed a significant increase in theta2 and alpha1. There were also increases in theta1 observed in the 15–24 and 35–44 age groups; When compared to MR controls, similar differences in alpha1 and alpha2 were observed in older adults.
[27]	- 33 younger adults with DS (20–35 years); - 12 older adults with DS (36–55 years); - 20 younger healthy adults as a controls (age 20–35 years); - 20 older healthy adults as a controls (age 36–55 years).	The relative and absolute power of delta, theta, alpha (8–11.9 Hz), and beta bands, along with their topographical distribution were investigated. Results were deemed abnormal if they deviated beyond the mean normal values ± 2.5 standard deviations.	+ Dementia was assessed using criteria of Shapiro, some patients had dementia; + Presence of a control groups; - There was no information whether DS was confirmed by genetic tests; * Lack of access to information about the genetic confirmation of DS, the presence of coexisting neurological and psychiatric diseases, epileptic seizures and psychoactive drugs taken.	Results indicated abnormal EEGs in 73% of young adults with DS and 92% of older adults with DS; Both age groups with DS displayed heightened power in delta (both relative and absolute, predominantly in centro-anterior regions), theta (both absolute and relative, mainly in centro-posterior regions), and beta (both absolute and relative, mostly in parieto-temporal regions), coupled with a decrease in alpha; The decrease in alpha was statistically significant only in the posterior regions among older adults; When comparing older adults with DS to young adults with DS, only the older adults exhibited a significant increase in theta power; Overall, participants with DS demonstrated a significantly slower alpha peak rhythm compared to controls (9.2 ± 1 Hz vs. 9.7 ± 0.3 Hz), with no correlation between alpha peak frequency and age in either group; In participants with DS, alpha peak frequency positively correlated with scores on neuropsychological tests such as Raven Colour Matrices, Rivermean Behavioural Memory Test, and Tolken Test; The severity of cognitive impairment was associated with a higher prevalence of abnormalities, including decreased alpha power, increased theta in posterior regions, and decreased delta across the scalp. This pattern was more pronounced in participants with dementia.
[28]	- 45 adults with DS (mean age 30.6 years), subdivided into 33 younger adults (age 20–45 years) and 12 older adults (age 36–56 years); - 40 healthy control subjects (20 younger adults (age 20–35 years) and 20 older adults (age 36–55 years).	Resting-state analysis involved computing the relative and absolute power of delta, theta, alpha (8–11.9 Hz), and beta. The topographic distribution of each band was examined using bidimensional maps with rectangular linear interpolation. Results were deemed abnormal if they deviated beyond the mean values of the control group ± 2.5 standard deviations.	+ Participants were divided into two groups: younger adults and older adults; + Dementia was diagnosed through clinical assessment using the Shapiro criteria and cognitive test outcomes; + Presence of a control group; * Lack of access to information about the genetic confirmation of DS, the presence of coexisting neurological and psychiatric diseases, epileptic seizures and psychoactive drugs taken.	Results showed that 73% of young adults and 92% of older adults had categorised EEG readings as “abnormal”; In the younger adults’ group, there was a notable increase in delta (75%), theta (37%), and beta (37%) power, along with a 58% decrease in alpha power; The older adults’ group displayed a significant increase in delta (54%), theta (91%), and beta (36%) power, coupled with a 45% decrease in alpha power; Topographically, there was a significant rise in delta power in centro-anterior and parieto-temporal regions, and a noteworthy decrease in alpha power over posterior regions in the older adults group; There was a significant increase in absolute and relative theta power over centro-posterior regions in the older adults group; Cognitive impairment correlated with a decrease in alpha power, an increase in theta power (particularly in posterior regions), and an increase in delta power across the scalp. This pattern was more pronounced in participants with DS and dementia; Individuals with DS without EEG abnormalities performed significantly better on most measures compared to those with abnormalities.
[29]	- 25 adults with DS (mean age 38 years, range 30–69 years); - 25 age-matched healthy controls (mean age 36 years, range 28–65 years).	There was calculated the average absolute power for each channel across the 1–30 Hz range with 1 Hz increments. Percent power was determined as a percentage value of a 1 Hz segment. The alpha frequency was identified as the frequency with the maximum power within the alpha band over occipital electrodes (O1 and O2). Absolute and relative power were assessed for traditional bands (alpha1&2 and beta1–3), and the absolute power of individually adjusted bands for theta and alpha 1–3 was also measured.	+ Patients with neurological and psychiatric disorders were excluded; + Patients taking psychoactive medications were excluded; + Presence of the control group; - No genetic confirmation of the presence of DS or its subtypes; - Wide age range of examined patients and no division into groups of younger and older adults; - No assessment for cognitive decline and dementia; - Patients were not examined for the occurrence of epileptic seizures.	A significant decrease in alpha frequency in DS compared to controls; Positive correlations were observed between WAIS-total and MMSE scores with alpha frequency in DS; Negative correlations were noted between WAIS and RBM scores and occipital alpha power in DS; In the analysis of absolute power in fixed bands, DS exhibited significantly higher absolute power and increased CSD in theta, alpha1, and beta1 compared to controls; Negative correlations were identified between WAIS scores and CSDs in the right frontal lobe and right PCC, while RBM scores showed a negative correlation with CSD in right BA9. A negative correlation was found between absolute occipital power and cognitive test performance using 1Hz bands; Analysing absolute power in IAF-adjusted bands, DS showed a shift of maximum alpha power to the lower alpha2 band, unlike the upper alpha band in the control group; DS demonstrated significantly increased CDS in all individually adjusted bands compared to controls; WAIS scores exhibited a negative correlation with CSD in right BA9 in the theta band; DS, compared to controls, displayed significantly higher absolute power in theta and alpha bands; Concerning relative power, the maximum value of alpha was found in alpha1 in DS and alpha2 in controls. A notable difference between groups was observed in alpha2, indicating decreased CSD in the cingulate cortex in DS; No correlations were found between regions of decreased CSD in alpha2 and cognitive scores; The average value for relative alpha2 and its occipital values correlated with cognitive scores, showing a negative correlation with power at 7–8 Hz and a positive correlation at 11–12 Hz.
[30]	- 36 adults with DS (mean age 30.9 years, range 16–56).	Spectral estimates were acquired through multitaper analysis for each channel, employing 2-s windows with a time resolution of 400 ms, steps of 50 ms, and a 3 dB bandwidth. Averaging was performed within and across time windows for each subject, and scalp maps were generated for statistical parametric mapping results. Spatial smoothing was applied to minimize spatio-anatomical differences between participants. SPMs were family-wise error corrected at *p* = 0.05. Mean scalp maps for theta (5 Hz) and alpha (8 Hz) power were created to analyse their regional distribution and locate the maximum power in each band. Linear regression was then employed to explore the relationship between raw KBIT-2 scores and both alpha peak amplitude and alpha peak frequency derived from regional electrode averages. Additionally, a scalp-wide SPM for power in the combined theta-alpha range (4–13 Hz) was generated. Regression analysis identified significant associations between raw KBIT-2 scores and theta-alpha power across the scalp, with family-wise error correction applied. This comprehensive analysis was conducted using a general linear model.	+ All participants had genetically confirmed trisomy 21; + Patients with confirmed absence of dementia; + Patients with mental disorders were excluded; - No control group; - Patients were not examined for the occurrence of epileptic seizures.	The study’s findings indicated a predominant theta activity in frontal regions and alpha activity in posterior regions for all participants based on average scalp maps and mean power spectra; Linear regression analysis demonstrated a significant association between raw KBIT-2 scores and peak alpha amplitude specifically in the frontal region; The correlation between peak alpha amplitude at the occipital region and raw KBIT-2 scores did not reach statistical significance; There was no significant association found between peak alpha frequency and KBIT-2 scores for either frontal or occipital electrodes The whole-scalp SPM analysis of spectral estimates revealed clusters with significant positive correlations between estimated general cognitive ability (raw KBIT-2 scores) and power in the 4–13 Hz range.

Note: In the column, Controls for confounders” a plus (+) means a factor that has a positive effect on the control of confounding factors, a minus (-) means a factor that has a negative effect, and an asterisk (*) means a lack of access to the study data.

## Data Availability

No new data were created or analysed in this study. Data sharing is not applicable to this article.

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
