# Peer review of "EEG in Down Syndrome—A Review and Insights into Potential Neural Mechanisms"

_brainsci, 2024, doi:10.3390/brainsci14020136_

Round 1

Reviewer 1 Report

Comments and Suggestions for Authors

Thank you for inviting me to review this manuscript. The authors provide an overview of EEG studies in Down syndrome. They summarise the studies and attempt to connect findings to potential underlying neural differences in this population, focusing on alpha, delta and theta frequencies. 

This is certainly a worthwhile review, and the authors appear to have closely examined the included studies. Relating findings to potential underlying neuropathologies is also very valuable. However, there appears to be some key studies missing (those which I have identified are provided at the end). I believe the chosen search terms should include “ERP” (if applicable, see below) and “trisomy 21”. I also believe more studies will be identified by including the following terms: “alpha”, “delta”, “beta”, “theta”, gamma”, “p300”, “oddball”. 

It would be beneficial for the authors to clarify in the Introduction section the differences between evoked (e.g. EPR studies) and non-evoked (e.g. resting-state) studies. It appears that the authors have focused on resting-state studies – this should be clarified, and the rationale for this decision explained (if applicable). The authors should also clarify in the introduction the differences between EEG and QEEG. Additionally it would be worthwhile clarifying differences between relative vs absolute power, and highlighting within the discussed studies which the authors measured.

Furthermore, there are several key issues with interpreting EEG research in DS that should be addressed in this review. This includes potential issues with comparing DS to typically-developing control subjects (e.g. controlling for differences in brain volume, education, medical comorbidities etc). There are also important issues comparing within DS, which should be highlighted – including with comparing between different age groups (i.e. comparing children with DS to older adults with DS who are likely to have significant amyloid burden).

With regard to the evaluated studies, it would be useful to add whether participants had genetically confirmed trisomy 21 (misdiagnosis may be an issue with earlier studies, and additionally there are partial/mosaic forms of DS which may confound results). It would also be good to confirm whether cognitive decline was checked for, and if so how (e.g. CAMDEX), and whether there were any attempts to control for this in the study.

Further details on analysis methods would be very useful. For example, it is not clear from study [12] (line 112) how alpha was analysed (it sounds like by eye) and how “signs of dementia” were determined. I recommend including a table of the studies summarising the main methods (i.e. age group, EEG methods, controls for confounders etc) and main findings for each included study. It would also help the Results section if studies were grouped in some way – such as by child/adult/older adults, study type (e.g. longitudinal), analysis methods, and/or key findings.

As for the Discussion section, it would be useful to discuss possible sources of any inconsistencies in findings (e.g. were the populations or analysis methods different?). It should also be explained why gamma is not relevant. Many previous studies do not measure gamma, as this is difficult in DS due to excessive muscle artifacts. If this is the case, it should be mentioned. Likewise, if there is another reason why gamma was not discussed this should be explained.

There are many opportunities to be more specific in the Discussion. For example:

  • “future research utilising bigger cohorts and standardised methodology is essential”. Based on your review and suggestions by previous authors, what “standardised” methodology would you recommend? This would be useful to include.
  • “aiding in early detection and targeted interventions” – you could discuss the opportunities in this area, such as the rise of at-home wearable EEG for monitoring. 
  • “medications that impact cholinergic pathways” – have any been trialled in DS?
  • This insight is pivotal for tailoring interventions that address the diverse needs of individuals across different life stages” – suggests personalised medicine, but not discussed.

Further comments on specific lines: 

Line 352 – "Elevated slow-wave activity across various brain regions in DS patients suggests 

potential abnormalities in neuronal synchronisation and communication.” This is vague and could apply to any EEG abnormalities. 

Line 464 – “The hippocampus, crucial for memory formation and spatial navigation [51, 52], significantly contributes to these oscillations.” I’m not sure it’s true that hippocampal theta significantly contributes to scalp theta. Please provide a reference. 

Line 488 – "may be one reason why people with this condition face cognitive difficulties” – consider that EEG abnormalities in DS could be compensatory, not primary pathological (e.g. as discussed in Hamburg et al. 2019).

Line 492 – “Longitudinal studies have been instrumental in unveiling the neurodevelopmental trajectory of DS and understanding how EEG patterns change with age.” – a summary diagram of how EEG patterns change with age in DS, according to your identified studies, would be useful. 

Line 349 – “normally developing people”. It is generally considered more respectful with discussing DS to use the terms typical/atypical, rather than normal/abnormal.

Studies that appear missing:

Salamy, Alvarez, Peeke (1990). Neurometric evaluation in down syndrome individuals:  possible implications for dual diagnosis.

Ono, Yoshida et al. (1992). Age related changes in occipital alpha rhythm of adults with  Down syndrome.

Locatelli, Fornara et al (1996). QEEG in adult patients with Down syndrome.

Medaglini, Locatelli et al. (1997). P300 and EEG mapping in Down’s syndrome.

Velikova, Magnani et al. (2011).  Cognitive impairment and EEG background activity in adults with Down syndrome: A Topographic Study.

Hamburg, Rosch et al. (2019). Dynamic Causal Modeling of the Relationship between Cognition and Theta–alpha Oscillations in Adults with Down Syndrome.

Saini, Masina et al. (2023). The mismatch negativity as an index of cognitive abilities in adults with Down syndrome.

Author Response

Dear Reviewer,

I would like to thank you very much for reviewing my article. Your valuable comments contributed to significantly improving the quality of this manuscript. Below are the answers to your suggestions:

Ad. 1-I added all missing studies.

Ad. 2- In the introduction, I added information about the differences between EEG methods (ERP and rs-EEG). The decision to include EEG tests was justified. I wrote the difference between EEG and QEEG. I added information about the difference between relative and absolute power. In the discussion of the research, I added information (if available) about the type of power measured.

Ad. 3 - Now all sources of inconsistency between studies are included in the subsection “4.2. Sources of inconsistencies in results and prospects for avoiding them in future research. Recommendations for future research on how to avoid these inconsistencies are also provided.

Ad. 4 - I added information in the discussed studies whether DS was genetically confirmed. In addition, I added information on whether the studies measured I AD cognitive decline (the method of measurement was also provided), where available.

Ad. 5 – signal analysis methods are given (in the table). A table with an overview of the research has been added. The “results” section is divided into two subsections: “Longitudinal studies” and “non-longitudinal studies”.

Ad. 6. In the discussion, as I mentioned earlier, there is a subsection “4.2. Sources of inconsistencies in results and prospects for avoiding them in future research", which discusses possible sources of inconsistencies in research. It was mentioned that many studies did not measure gamma activity and a justification was given (lines 661-664).

Ad 7. a - Description of methodological recommendations can be found in section “4.2”.

Ad 7. b- I wrote about the possibilities offered by portable EEG to perform at home and about the possibility of using neurofeedback therapy aimed at deviations from EEG activity (lines 631-636).

Ad. 7. c – I wrote about the fact that cholinergic drugs are not usually tested in people with DS, but I cited one study that did use them (lines 717-718).

Ad. 7. d - I mentioned personalized medicine (lines 658-660), which is discussed in more detail on lines 710-718.

Ad. 8 - Line 352 – now lines 607-608 – I removed this sentence.

Ad. 9 - Line 464. I added a reference that does not contradict the claim that hippocampal theta can influence scalp theta (ref. 70).

Ad. 10 – Line 488 “may be one reason why people with this condition face cognitive difficulties” – consider that EEG abnormalities in DS could be compensatory, not primary pathological “. This is a hypothesis, but further research is required to confirm it. Epilepsy is very common in people with DS, so I wonder if these mechanisms exist and whether they really reduce the incidence of epilepsy in people with DS.

Ad. 11 – Line 492. A diagram is presented showing how deviations in EEG activity change with age in people with DS.

Ad. 12 – Line 349. Changed “normally” to “typically”.

Reviewer 2 Report

Comments and Suggestions for Authors

Thank you for the opportunity to review this fascinating and important manuscript on EEG patterns in DS. The authors did a commendable job in summarizing the existing research, and explaining the neurophysiological relevance of the findings. I believe the paper should be accepted with minor revisions. My comments are below:

1) The summaries on alpha, theta, delta, and beta within the discussion should be presented in the introduction instead. These summaries provide useful information but they should be condensed to some degree.

2) The authors should discuss further the mixed findings for the beta frequency. Could this be a result of co-occurring anxiety or neuroinflammation in some cases.

3) The authors should offer some insight as to how EEG can guide the selection of certain medications or other interventions for DS. Have any of these studies been conducted?

Author Response

Dear Reviewer,

I would like to thank you very much for reviewing my article. Your valuable comments contributed to significantly improving the quality of this manuscript. Below are the answers to your suggestions:

Ad. 1. Dear Reviewer, we would like to leave the discussion of individual waves as is - in the discussion. We did this for two reasons. First, we discuss the waves that are relevant to EEG research in DS. If we discussed the sources of the waves and their general features in the introduction, we could not write this in the discussion and readers might miss the connection of the general features of these waves with their neural correlates in DS. Secondly, this review is specialized and concerns EEG in a specific disorder – DS. I think that readers interested in reading more or less know the characteristics of waves occurring in EEG. And if not, there are plenty of comprehensive reviews discussing the characteristics of all waves.

Ad. 2. We discussed the differences in beta activity findings and gave possible reasons (lines 665-674).

Ad. 3. We presented what interventions can be used when we know the abnormal EEG activity - cholinergic drugs and neurofeedback therapy. Lines 632-636, 710-718.

Round 2

Reviewer 1 Report

Comments and Suggestions for Authors

Thank you very much for inviting me to review this revised manuscript. I felt the paper was much improved. I think the comprehensive section on sources of inconsistencies will be especially useful for informing the design of future experimental studies.

I am still unsure why "trisomy 21" was not included as a search term, as this is the medical term for Down syndrome. If you did search for this but didn't find any additional papers, it should be included in the list of search terms used. If not, I would advise running the search again with the addition of this term.

I think it is very useful that you comment on whether studies were eyes-open or eyes-closed, as this can have a significant impact on EEG recordings. However, this is not explained in the paper. I think adding a few sentences in the introduction about why eyes open or closed is important to consider will improve the paper.

Finally, I didn't understand what was meant by the sentence "As a result, the test sample will be homogenous" in the context of choosing to focus on resting-state EEG studies. I also didn't understand what was meant by "knowing the wrong frequency band" (line 615). It would be beneficial to reword these sentences to clarify your intended points. 

Author Response

Dear Reviewer,

Once again, thank you very much for carrying out the minor revision. I followed your comments, as described below.

Ad. 1 We included the phrase "trisomy 21" in the study inclusion process and performed the search, but this did not affect the acquisition of additional studies. Identified studies examined patients with DS and epilepsy.

Ad. 2 We have added information about the EEG measurement method with eyes open and closed and what effect it has on the EEG image (lines 89-95). Furthermore, we wrote that the included studies measured EEG mainly with eyes open, and suggested that future studies should test both conditions, especially eyes closed, due to the predominant alpha activity, which is our primary marker of cognitive dysfunction in DS (lines 832-836).

Ad. 3. The mentioned sentences are corrected, I hope that now their meaning is clear (first sentence lines 78-79, second sentence line 633).